# Very Young Child Survivors’ Perceptions of Their Father’s Suicide: Exploring Bibliotherapy as Postvention Support

**DOI:** 10.3390/ijerph182111384

**Published:** 2021-10-29

**Authors:** Cortland Watson, Elizabeth A. Cutrer-Párraga, Melissa Heath, Erica E. Miller, Terrell A. Young, Suzanne Wilson

**Affiliations:** 1Maricopa Unified School District 1, 44150 W Maricopa-Casa Grande Hwy, Maricopa, AZ 85138, USA; cortlandwatson10@gmail.com; 2Counseling Psychology and Special Education, 340 MCKB, Brigham Young University, Provo, UT 84602, USA; melissaheath3@gmail.com; 3BYU-Marriott School of Business, N. Eldon Tanner Building (TNRB) Campus, Provo, UT 84602, USA; ericaellsworthmiller@gmail.com; 4Teacher Education, 205 F, McKay School of Education, Brigham Young University, Provo, UT 84602, USA; terry_young@byu.edu; 5Davis School District, 45 E. State St., P.O. Box 588, Farmington, UT 84025, USA; suzynwilson@gmail.com

**Keywords:** father’s suicide, child survivor, suicide prevention, grief, bibliotherapy, communication, tasks of grief

## Abstract

Each year in the United States, 7000 to 30,000 children experience their parent’s suicide. Due to the stigma associated with suicide, feelings of guilt, and intense grief, surviving family members avoid talking about suicide. Over time, children struggle with confusion and intense emotions associated with their parent’s suicide. In this study, seven adults, who reported being younger than six years old at the time of their father’s suicide, participated in individual semi-structured interviews. Participants’ responses highlight the challenges that young children face due to limited memories of their deceased parent. Interviews concluded with an opportunity for participants to review and express their impressions of 10 children’s picture books. Participants offered impressions about how these books may or may not be helpful in supporting young child survivors. Implications for applied practice include considering how children’s literature may open communication and assist children in navigating Worden’s tasks of grief: (a) accepting the reality of their parent’s death; (b) facing the grief and pain; (c) adapting to life changes due to their father’s suicide, in particular adapting to altered family relationships; and (d) building memories of the deceased loved one, when possible, to ensure healthy attachment to the deceased parent. Participants’ insights provide considerations for selecting children’s literature for bibliotherapy. Due to young child survivors’ increased risk for attempting and completing suicide, supporting child survivors of parent suicide not only addresses postvention needs but aligns with suicide prevention.

## 1. Introduction

The World Health Organization (WHO) identifies suicide as a public health priority and suicide prevention as a “global imperative” [1] (p. 7). They estimate that, worldwide, approximately 700,000 individuals die from suicide each year [1]. Placing this number in context with other types of death, the WHO reports that more individuals die from suicide each year than from homicide and war, breast cancer, or malaria. In the United States, suicide is the tenth leading cause of death, accounting for approximately 48,000 deaths each year [2]. On a national level, information posted on the U.S. CDC website states: “Suicide is a serious public health problem that can have lasting harmful effects on individuals, families, and communities” [2].

For survivors—family members and friends of the person who completed suicide—the stigma surrounding suicide amplifies and complicates their grieving process, impeding opportunities to talk about the death [3,4,5,6,7]. This intensifies survivors’ isolation and alienation [4,8]. Tragically, many survivors suffer in silence and with limited emotional support [4,8,9,10,11]. Although research is limited, Wilson and Marshall indicate that less than half of suicide survivors seek professional assistance, and, of those, approximately 40% reported that the assistance was helpful [11].

### 1.1. Child Survivors of Parent Suicide

Of all traumatic childhood events, the death of a parent is one of the most stressful [12,13,14]. For surviving children, a parent’s death triggers both short- and long-term physical, emotional, behavioral, and mental health challenges [15,16,17]. In addition to the loss of the parent, the associated cause of death also has implications for surviving children [15,16]. In particular, the death of a parent by suicide is especially traumatic for surviving children [17,18].

Researchers estimate that the number of children in the United States annually who experience their parent’s suicide ranges from 7000 to as high as 30,000 [19,20]. These child survivors experience more complicated grief compared to children bereaved by a parent’s non-suicidal death [18,21,22]. When compared to typically developing peers, child survivors of a parent’s suicide (CSoPS) experience higher levels of depression, anxiety, and anger [23,24]. They often struggle with peer relationships, academic achievement, and across the years struggle with interpersonal relationships and have difficulties attaining job satisfaction and sustaining employment [25,26]. Additionally, one of the most worrisome outcomes for CSoPS is that they are three to four times more likely to attempt and ultimately complete suicide compared to youth from families with two living parents [27,28,29,30]. As such, CSoPS are identified as a highly vulnerable population [17,18,20]. Even though there is a great need to support child survivors’ immediate and ongoing needs, research examining how to assist this young population is very limited [17,18,31].

In particular, immediate and ongoing needs of CSoPS include supporting and encouraging children to ask questions, talking about the parent’s death, and helping children understand *why* the parent chose to die by suicide [7,20,32,33]. However, research indicates that this communication is often very limited and that formal and informal support are difficult for child survivors to access [9,18,19,20,34]. For CSoPS, a lack of communication and unmet emotional needs complicate their grieving process [7,19,20,34].

To help children adaptively cope with and manage strong emotions associated with a parent’s suicide, Schreiber et al. and Montgomery and Coale emphasize the need to present information about death and suicide on a level that is developmentally appropriate for the child [18,35]. Ultimately, the choice should not be *whether or not* to talk about the parent’s suicide, but *how* to talk about it and how to encourage honest conversation [33,35].

### 1.2. Young Children’s Adjustment following a Parent’s Suicide

When investigating children’s behavior and socio-emotional development following a parent’s suicide, researchers offer mixed results. Kuramato et al. noted that children who experienced their mother’s suicide were at a higher risk for suicide attempts resulting in hospitalization than were survivors of a mother’s death from unintentional injury [28]. However, Tsuchiya et al. and Geulayov et al. noted that in comparison to adolescents and young adults, younger children exposed to maternal suicide appeared to suffer greater ill effects across time [36,37].

Kuramoto et al. noted that adolescents are especially vulnerable to suicidal ideation and completion of suicide in the first two years following their parent’s suicide, with risk declining after that point in time [28]. Although minimal research has been conducted with very young CSoPS (under the age of six when a parent completes suicide), Kuramoto et al. noted that young CSoPS, not adolescents and young adults, have the highest rates of suicide across their life span [28]. Although Kuramato et al. suggest that grief stemming from a mother’s suicide may differ from grief from a father’s suicide, this difference did not significantly alter the risk trajectory for attempting and completing suicide [28].

Although the importance of the mother’s mental health has continuously been identified as a critical factor in children’s development, the importance of the father’s influence on children’s mental health and behavior cannot be ignored [37,38,39]. Emphasizing the important role fathers play in their children’s physical and mental well-being, Azuine and Singh note that actively involving fathers in their children’s lives “may provide a potential opportunity to reduce mental and emotional health problems among children” [38] (p. 495).

### 1.3. Tasks of Grief

Although each person’s grief is unique, there are commonalities that define the grieving process. Initially, linear stages of grief were proposed by Kübler-Ross [40]. However, following Worden’s study of children grieving their parent’s death (not specific to death by suicide), he identified four *tasks* of grief: (a) accepting the reality of death; (b) facing the emotional pain of grief; (c) adjusting to the changes after a loved one’s death; and (d) remembering and memorializing the death and life of the deceased person [14]. Worden stated that the tasks of grief are not sequential in nature, nor is there an endpoint when one’s grief is completed [14]. The tasks of grief reemerge across time as the individual remembers the deceased loved one and experiences major life events without their loved one’s physical presence. Additionally, as children develop and mature, their grief is redefined. With maturity comes an increase in life experience and a growing understanding of the nature of death. With sufficient support, children’s grief becomes a part of their daily lives, and, over time, feelings associated with grief decrease in intensity [14]. However, Wolfelt warned that if “…children are not compassionately companioned through their complicated mourning journeys, they are at risk for behavioral and emotional problems” [41] (p. 655). Caring adults in all areas of a child’s life have opportunities to provide compassionate support.

### 1.4. Bibliotherapy to Support Young CSoPS

Bibliotherapy, a term first used by Crothers in 1916, relies on the power of stories to help individuals cope with and process difficult experiences [42,43,44,45,46,47,48,49]. In general, practitioners support the use of bibliotherapy as a developmentally appropriate counseling intervention strategy to open communication about death and to support children who struggle coping with their grief [43,44,45].

The type of books used for bibliotherapy may include fiction, non-fiction, poetry, and children’s literature. Stories are selected that will best support children in coping with their loss [43,46]. Stories provide different perspectives, support children’s socio-emotional growth, and normalize experiences, helping children know that they are not alone and that others have experienced similar situations [42,47,48]. Additionally, Berns indicates that through the safety of storybook characters, children may more freely share feelings when addressing challenging situations [44].

Although a few internet websites (e.g., Brigham Young University Building Social Skills with Books (https://education.byu.edu/buildingsocialskills/managing-grief, accessed on 27 October 2021; the Dougy Center (https://www.dougy.org/resources/audience/kids?how=&who=&type=&age=0-6, accessed on 27 October 2021), and the American Psychological Association (https://www.maginationpressfamily.org/stress-anxiety-in-kids/tag/grief/, accessed on 27 October 2021) list young children’s books about grief and loss or suicide, minimal research has been conducted with this type of support for very young children [6]. We also note that several survivors of suicide websites provide suggestions for young children’s grief- and suicide-themed books (e.g., various sites sponsored by Survivors of Suicide Loss (https://www.sosmadison.com/books/helping-children-and-teens-cope-with-a-suicide, accessed on 27 October 2021; and https://www.soslsd.org/bookstore/children/, accessed on 27 October 2021).

### 1.5. Focus of This Study

Additional research is needed to inform mental health practitioners, teachers, and parents/caregivers about providing support to young CSoPS (less than six years old). Caring adults need specific strategies to facilitate communication about the parent’s suicide [6,7]. Noting these needs, the primary purpose of this study was to explore how adults who, as very young children, lost a father to suicide thereafter perceived their father’s suicide, and obtain their recommendations regarding children’s literature (picture books) that may or may not support young CSoPS. More specifically, the goal of this study was to lay the foundations for gathering information that will inform interventions, including potential bibliotherapy-based intervention [7], to support young CSoPS. Noting that young children rarely receive professional mental health services [50], we also hope this study will provide some guidance for parents and caregivers who need strategies to open communication with young survivors [7].

## 2. Methods

Prior to conducting this research, the proposed study was approved by the sponsoring university’s institutional review board. Documentation of this approval is available upon request. Each participant signed an informed consent form prior to being interviewed. Given the vulnerability of participants, as well as the deeply sensitive and personal nature of surviving a parent’s suicide, each participant was offered contact information (email address and cell phone number) of a licensed psychologist who had experience working with parents and children following traumatic events. If sought, counseling was offered at no cost to participants.

### 2.1. Methodological Framework Considerations

The methodological framework for this study comprises certain philosophical considerations. That is, we presupposed an associative ontological perspective. We assumed knowledge is constructed, enlarged, and maintained through relationships rather than through independent artifacts [51,52]. Hence, our decision to look at very young children and their shared experiences surviving a parent’s suicide.

Second, because this study’s foci were intended for CSoPS and feedback was sought to inform caregivers’ postvention, we concurred that the study would also be grounded in Worden’s philosophy of the four tasks of grief [14]. We did not know at the outset if or how participants’ experiences would align with Worden’s philosophy. However, we opted for a more postmodern philosophical foundation that would allow us to simultaneously understand the relational components of very young CSoPS experiences while recognizing the individual narrative within each child’s trajectory with Worden’s work.

Third, in planning this study, we recognized the structural incertitude and the benefits of seeking to understand the retrospective experiences of participants from a vulnerable group (e.g., CSoPS). This study’s data are sourced from adult participants’ retrospective memories of their experiences following their parent’s suicide. We dispute the criticism that this type of memory work is too subjective an authority for social science research [53]. Rather, we believe that retrospective interviews by adults tend to consist of greater depth and emotional poignancy than similar work conducted with children, given that adults tend to include more accurate representations of past family events and can help establish an absent voice for vulnerable or silenced populations, such as CSoPS [53,54,55]. Accordingly, we incorporated the following principles of rigor for retrospective work in our study by: (a) developing research questions that focused on each participant’s experiences specific to the parent’s suicide rather than a life in its entirety; and (b) comparing participants’ experiences of parent suicide with participants’ present thoughts about children’s books that may be helpful (or unhelpful) to CSoPS. Consequently, we realized that child survivors’ perceptions across time presented targets for supportive intervention, whether consistent or inconsistent [55].

Further, because this research uniquely offers the input of seven adults (CSoPS; see participants) who, as young children, experienced their father’s suicide, we have some initial groundwork on which to build a basis for proposed interventions that could potentially include bibliotherapy. Here is our logical reasoning for delving into this line of research: (1) we know that CSoPS are a vulnerable group and at increased risk of suicide attempts and completions [10,21,22,23,36]; (2) we know that research and proposed interventions for CSoPS are almost nonexistent; (3) we know that the vast majority of CSoPS will not have formal counseling (for a variety of reasons) [10,21,22,23,36]; (4) we know Worden’s tasks of grief and how to address these tasks for child survivors of parent death (not specific to suicide), but prior research has not linked these tasks with the grief of CSoPS [14]; (5) we know that bibliotherapy is utilized by counselors to address a variety of developmental challenges, including grief and loss—and we know that parents and caregivers could access children’s picture books [4,46,47,48]; (6) we know that qualitative research gives CSoPS the opportunity to tell their story [56]; and (7) we respect each participant’s life experiences and believe that participants’ insights offer some guidance for moving forward with offering tentative recommendations for using bibliotherapy with this vulnerable group [46]. With these considerations and groundwork in place, we moved forward with the study.

### 2.2. Research Design and Recruitment

#### 2.2.1. Case Study

A collective case study design was utilized for this study [57]. Case studies have been appropriately used across the social sciences to examine the context of real-life phenomena [58,59,60]. Qualitative methodological experts have suggested that case study design is typically chosen to increase our understanding of phenomena that are unusual, uncommon, or unique (e.g., adults who, as very young children, experienced their father’s suicide) in order to answer “how” questions [61,62]. The current study was especially well suited for this design in that the researchers sought to understand how each participant made sense of surviving a parent’s death by suicide. Capturing the participants’ perceptions of change over time in seeking to understand the long-term effects of surviving paternal suicide was another benefit of this study’s type of methodology.

As is common with case study research, this study incorporates data from multiple participants [63]. Data included interviews and field observations taken as the participants talked about their experiences and then offered their insights about the potential for children’s literature to support (or fail to support) CSoPS. It should be noted that this case study was exploratory in nature. Given the uniquely difficult and potential triggering nature of the topic, the research team was purposefully nondirective as we sought participants’ perceptions of children’s literacy books that may or may not support CSoPS.

#### 2.2.2. Participants

Qualitative participants are considered the experts as they share their perceptions and as researchers attempt to understand those perceptions [56] (p. 6). Thus, identifying participants who can inform significant insights and viewpoints relative to the research phenomenon is of critical importance to qualitative inquiry [64]. In addition to choosing instructive participants, an appropriate sample size must be determined. However, while sample size in quantitative work pertains to statistical power, sample size in qualitative work pertains to sufficient information power [65]. Sufficient information power is determined by sample specificity, quality of dialogue, and analysis strategy. This means that the closer the match between participants’ experiences and the study phenomena, the greater the information power.

Participants were purposefully chosen for this study because they survived the trauma of paternal suicide and met the inclusion criteria. Specific inclusion criteria included the following: (a) the participant was younger than six years old when the father completed suicide; and (b) at the time of the interview the participant was older than 18. Seven participants from the larger data set met these criteria (*N* = 17) [7]. The participants’ demographic information is listed in Table 1. Three participants were female and four were male. At the time of their father’s death, three of the participants were five years old; one was four years old; one was three years old; one was one year old; and one was three months old.

Participants were recruited through invitations sent to bereavement groups, social media platforms, and suicide prevention groups. Additionally, information and invitations were posted on public and college library bulletin boards (within 50 miles of the sponsoring university). Because of the vulnerable and sensitive nature of the topic and participants, we did not approach individuals. Rather, interested participants contacted the researchers through email, if they were willing to participate. Following email contact, appointments were made for a face-to-face individual interview. After 17 participants expressed interest in participating, and after determining that the initial 17 individuals fit the sampling criteria, we informed individuals that the number of participants was sufficient and that the search was closed.

### 2.3. Data Collection and Analysis

#### 2.3.1. Interviews

Data collection took place through individual interviews and included observation notes taken during the interviews. Before beginning the interview, the participants filled out a brief demographic questionnaire. The initial interviews with the seven participants lasted an average of 60 min and took place in a conveniently located local library’s private meeting room.

#### 2.3.2. Observations of Interactions with the Books

Observations took place directly following the interviews as the participants were interacting with 10 children’s picture books. Each of the participants (as adults) were presented children’s books that counselors may share with children following a death or suicide. Books were selected based on a review of books recommended on grief-themed websites, reviews of books and recommendations available on Amazon, recommendations provided by individuals familiar with children’s literature (including counselors and librarians), and consultations with two bibliotherapy experts. Both of the bibliotherapy experts are nationally (United States) and internationally acclaimed PhD-level experts who have relied on bibliotherapy to support grieving children when working with young children, and who have researched, published, and presented on grief and bibliotherapy over the past 25 years.

Table 2 provides a summary and description of the available books. The interviews and observations were audio recorded and then transcribed verbatim. Nine of the books were specific to grief following the death of a loved one. Of these, four of the books were specific to suicide (one book about a mother’s suicide, two books about a father’s suicide, and one book about an uncle’s suicide). One book, *The Invisible String* (by Patricia Karst) is commonly recommended for therapeutic bibliotherapy for a variety of issues, including grief.

Each participant was asked to review the selection of children’s books and describe what would or would not be helpful for children who have experienced a parent’s suicide. The aim of this activity (reviewing the books) was to help provide feedback for selecting books that would specifically help a bereaved child understand suicide and process their emotions following such a tragic event. The interviewer observed the participant looking through and commenting on several books. Participants were also observed as they provided explanations as to why they chose specific books and whether or not they felt that the books would be helpful for young CSoPS.

After the interviews and observations were completed, audiotaped interviews were transcribed. Each participant was provided with their transcribed interview for their review. Participants were offered the opportunity to clarify points in the transcript, or to revise their comments, if needed. Following approval from the participants (member checking), the transcripts were de-identified and then uploaded to a secure password-protected server. For this current study, a second approval from the university institutional review board was granted to analyze the interview data specific to this study’s seven participants.

#### 2.3.3. Coding and Analyses

Transcriptions of each interview and observation notes were downloaded into a Word document which served as an organizational tool to analyze the data. The analysis was completed through a manual process of within-case and cross-case analysis.

#### 2.3.4. Within-Case Analyses

A priori codes were used during first-cycle within-case analyses. There were three major a priori codes used during first-cycle coding in order to organize the data. The three a priori codes were: (a) how the participant found out about the suicide, (b) how the participant reacted to the suicide, and (c) how the participant reacted to the books.

Next, emotive coding was used during the within-case analysis to explore other important aspects of the individual experiences of each participant. Emotive coding is a process used to examine evidence of organizing and revealing interpretive and personal meaning of data, such as what a participant might be experiencing in their mind [66,67]. This type of coding has been deemed appropriate when exploring intrapersonal and interpersonal experiences that help individuals with decision making and making judgements. This type of coding has also been found helpful when exploring the mood and tone of books [66,67].

During this stage of the analysis, process coding was also used. Process codes, often referred to as action codes that include gerunds, or -*ing* words, are used to help researchers understand change or shifts in thinking or behaviors over a period of time [67]. This type of coding was particularly helpful when exploring how participants reacted to the suicide.

We then compared and contrasted the portraits from each participant and their perspectives on the books. Each portrait is representative of a participant. This allowed researchers to notice differences and similarities between each participant. See Table 3 for a sample participant portrait.

#### 2.3.5. Cross-Case Analysis

The second phase of data analysis included the researchers creating a table to organize the data across all participant portraits. The next step of the analysis process included the research team engaging in multiple data review rounds to make meaning of and reduce the codes from across the participants. The purpose was to condense the code lists into overarching themes that would robustly describe participants’ experiences. The resulting six overarching themes included:Theme 1. How the participants found out about the suicide from othersTheme 2. On their own: How participants found out about the suicideTheme 3. Elements that activated the healing processTheme 4. Elements that prolonged the healing processTheme 5. Books participants found helpfulTheme 6. Books participants found unhelpful

The use of in-depth fusion of coding from within and then across coding strategies resulted in a more in-depth understanding of each participant’s experiences.

### 2.4. Trustworthiness

Indicators of rigor that promote the trustworthiness of data in qualitative studies include standards of credibility, transferability, dependability, and confirmability [68,69].

For those who are more familiar with quantitative methodologies, these indicators help translate quality study traits. For instance, an indicator for rigor in quantitative studies concerns itself with statistical generalization to systematically apply study results to homogenous populations. Rather than seeking to generalize study results to homogenous populations, a comparable indicator for qualitative work is consideration for the resonance or the *naturalistic generalization* of the work [70]. Naturalistic generalization speaks to transferability and includes a process in which readers intuitively apply a study’s findings to a comparable context [70].

In order to ensure trustworthiness in this study, triangulation, peer review, member checking, thick description, external auditing, and researcher reflexivity were included [68]. These processes of trustworthiness are more fully described as follows. Using multiple views of the same phenomena (multiple participants, multiple observations, and interviews) helped to triangulate and strengthen the findings. An outside expert member who is a doctoral-level, licensed psychologist with expertise in suicide postvention with CSoPS reviewed the findings. The purpose of the expert reviewer was not an attempt to legitimize the findings, but rather to consider the plausibility of the findings relative to current research and the professional literature related to CSoPS [71,72,73].

The team employed member checking during data collection. Given the vulnerable nature of the topic, participants were individually invited to review their own typed transcript and make corrections as needed to ensure accuracy. Thick descriptions of the context supported by evidence from the participants provided insights into individual perspectives and their personal meaning making of the phenomena [72].

To further ensure trustworthiness of the data, qualitative researchers are currently encouraged to practice reflexivity in order to strengthen rigor and to gain deeper interpretive access to the data [72,73,74]. Instead of seeking to bracket or diminish researcher roles in the inquiry process, or reporting on inter-rater and intra-rater reliabilities, reflexivity gives researchers the opportunity to acknowledge and explore their role in the data collection process [72,73,74]. The research team deliberately participated in several rounds of reflexivity to consider intersubjective dynamics between the researchers and the data in the current study and kept notes throughout the rounds [72,73,74].

## 3. Results

As indicated in the previous section, data from the in-depth interviews and observations revealed six overarching themes. Participants’ responses that support these themes are described in the following sections.

### 3.1. How Participants Found Out about the Suicide from Others

Because participants were so young at the time of the suicide, most had only vague memories of finding out about the suicide. For participants younger than five years old, their first memories of finding out about the suicide were from others. As a whole, they described finding out from others as a “traumatizing” and “confusing” experience. Participants shared how “others” would unexpectedly show up. Cody shared that his grandparents and uncles just showed up: “*But my grandparents just showed up and a couple of my uncles showed up as well.*” Similarly, Malinda said, “*The police just showed up at our house and my mom just, like, knew.*” Similarly, Justin shared his experience,

My uncles just showed up and then we started hunting for [my dad]. And then we went out to the barn, walked in, and I remember I was behind my uncle who was in front leading…. And I remember, feeling a kind of tension there. I think my uncle knew something was wrong.

After family members showed up, the participants describe the accompanying traumatic emotional reactions of nearby adults and family members. These emotional reactions included yelling, screaming, collapsing, and crying. For example, Malinda remembered,

I just have a distinct memory of, like, being at my grandparents’ house in the morning and seeing my mom, so very upset and, I just remember, like, that there were days when it was so hard, you know, where she didn’t want to, like, you know, keep living.

Justin related his uncle’s reaction when the uncle first spotted his father after the suicide. “*He got to the top of the stairs. I was just about to the top of the stairs myself and I remember him going ‘Oh shit!’ and yelling at me to run home now!*” Cody described his mother’s reaction: “*I heard my mom just collapse and she was screaming.*”

Next the participants described their reactions as children to the emotions of the adults as they found out about the suicide. Justin describes how frightened he was as he reacted to his uncle’s shouts to run home:

I remember running home and I kept thinking there must have been a monster in the barn. I had no idea. Then my mom left and the rest of us stayed home, and I remember we were just kind of sitting there, uh, in the living room, huddled up close together just scared, I had no idea what was going on.

Another participant, Danica, described how she reacted to her mother’s reaction after hearing the news of her father’s suicide.

Um, so she took me and my little sister into the shower and just started crying. She said it was so we could all cry and be wet at the same time. I didn’t really know what was going on and I was crying because she was crying.

The majority of participants also described the continued confusion as they observed the extreme emotional reactions of the adults. They reported not understanding what had happened. Cody said, “*One of my uncles, and he started talking to me about it, and he said, ‘Hey, your father passed away,’ but they didn’t say how it happened, what happened, anything like that.*” Malinda (four years old at the time of her father’s suicide) indicated that she was never told about the suicide until 14 years after the suicide. At this time, she was 18 years old. She reported,

I wasn’t really told the details of, like, how my father passed away, until very recently. I didn’t know it was actually this past summer where we, like, sat down, me and my [mom and stepdad] and my brother and I, and just, like, talked about it, what happened and [how he committed suicide].

### 3.2. On Their Own: How Participants Found Out about the Suicide

The participants each shared how they did not understand what had happened to their father or the specifics of how their father died. They describe acting as typical children prior to the suicide. For example, both Cody and Justin share how they were playing when others unexpectedly showed up and interrupted the playing. Cody remembered when adults showed up to tell him about his father’s death, “*I was young. I was playing in the gutter with some toys or whatever.*” Justin laughed as he reported, “*I remember going to their basement. I found a cookie monster toy that I loved and I was like ‘sweet!’ At the time, that’s all I cared about was that cookie monster toy.*”

Some of the participants talked about how they had a vague sense that something was wrong or missing. Cody shared, “*So my dad hadn’t come home for a little, for a couple days or a day or two, I can’t remember for how long it was.*” Malinda reported a similar experience:

Kind of just out of the blue, um, he just, like disappeared, um, just, like, didn’t come home from work, I think one day, and I do remember he was missing for a couple days.

It seemed it took some time, but each of the participants started asking questions about the father’s absence and about the death. For example, even though Justin’s mother had initially told him about his father’s death, he describes a process of wondering and trying to make sense of what happened to his father. At first, he did not know that his father died by suicide. Justin continued to ask his mother about his father’s death. He stated, “*Later on, a few years later, all I remember at first was asking, ‘Mom, who killed Daddy?’*” Danica described a similar experience:

So later, in like first grade I remember hearing…so I was like 6 or 7, so the next year…I remember hearing my mom and my grandma talking about Larry dying. And it was the first time that I remember hearing about his death and really connecting with it. I asked, ‘He’s dead, right?’

### 3.3. Elements That Activated the Healing Process

As participants disclosed ways in which they reacted to the news of their father’s suicide, they shared elements that helped to activate and as well as prolong the healing process. These elements included talking and overcoming, maintaining routines, accepting, and being faith filled. These elements are described in the following sections.

### 3.4. Talking and Overcoming

All of the participants described the healing effects of openly talking about the suicide, most often talking with a trusted family member or counselor. Cody related that he never talked about the suicide with anyone for 14 years. He went on to explain the healing effect of talking with a trusted counselor. He stated,

And as soon as I talked about it, like I said, I overcame it. It took a little while but eventually it happened, you know. I never went to counseling until [I was 19 years old]. And that’s when the healing process started to take place when I started going, going to counseling… I was 19, I think, roughly.

Malinda also commented on the helpfulness of talking to a counselor. She stated, “*I remember, like, going into, like, the counseling center and I always really liked going, like, I really, like, looked forward to it.*”

### 3.5. Maintaining Routines

Participants also related how helpful it was for them to keep regular routines as they moved forward in the healing process. Justin remembered going back to school:

It was a preschool. It was at this lady’s house up the street in her basement. She had a classroom. Going back to school must’ve helped to just normalize things because I felt pretty normal then.

### 3.6. Accepting

Participants shared experiences that helped them to accept the death of their father. Justin talked about developing the ability to share about his father’s death. He commented,

I remember I would anticipate it because every time you would meet new people, a friend’s parents or anything like that, you know, they’re inevitably going to start asking about family and all that so you kind of start to anticipate that question coming up and try to measure your response. But I got to where I would just say, ‘He killed himself.’ And that was it.

Malcolm shared how he came to accept his family life as his “new normal” after the suicide:

I had a mom and that’s my life. I had my sisters and yeah. It wasn’t abnormal to me even though I could probably step back and compare someone else’s family to mine and see that there’s differences, but, you know, at the time, and still to this day, I just feel like everything’s normal. It was just, that was life.”

### 3.7. Being Faith Filled

Another idea that helped participants in activating the healing process was the idea of faith. Cody shared his personal his insight:

The man upstairs changed me, God. And… I think having also, like, a belief in a higher power, belief in Jesus, or whoever those people may be to certain individuals, as they develop those beliefs in them they can seek healing through them, which I think is the ultimate goal. At least that’s how I healed completely, was through them.

Similar to Cody, Malinda, shared her beliefs:

I believe that we have, this living God who cares about us, and He’s not going to let us, be unhappy and the people who had to deal with those horrible things during their life, like, won’t have to feel the guilt for, like, their actions they took because, like, He understands perfectly. So I think that’s a huge factor in healing.

### 3.8. Elements That Prolonged the Healing Process

Participants also shared elements related to the suicide that prolonged the grieving and healing process. These elements included waiting to talk about the suicide; leaving, missing, and separating; acting out; and feeling paralyzed, conflicting feelings, and needing to fix it. These elements are further described in the following sections.

#### 3.8.1. Waiting to Talk about the Suicide

All of the participants discussed the difficulty of not being told about the suicide in ways that they could understand. Although several made similar comments, Cody’s comments sum up and represent the perception of needing to talk about the suicide.

I mean there’s these like un, unspoken rules that people follow. They are just like ‘Wait till they’re older to talk about it.’ So none of us [siblings] talked about it. We just kinda kept it on the back burner. [Looking back on it], I don’t think that’s wise. Yeah, so just not talking about it prolonged the healing process for each of us.

Danica added, “*I was so traumatized and so angry that I wasn’t told directly. That I had heard about it through eavesdropping. So like, there’s that whole thing!*”

#### 3.8.2. Leaving, Missing, and Separating

The participants described how they felt left by their fathers. Cody said, “*We’re like, what the heck we’re a family, why did you leave us?*” Jesse shared similar thoughts: 

I am missing something, and I can’t help but be reminded of it. So, I know it’s up to me to get over it but it’s just, I haven’t been able to. It is more his absence more than his death.

Malcolm reiterated the impact and subsequent feelings he experienced through the absence of his father. He commented,

So basically, I just grew up in a fatherless home. I didn’t have a father and my sisters didn’t really want to, um, hang out with the little boy. So, it shaped me. I mean, it would’ve been nice to have a father and to do this and that as a boy. So, I think it [the experience of growing up in a fatherless home] was extremely pivotal for me.

Danica also described what it was like to not have a father as she grew up. Yet, in her description, she seemed to indicate she was somehow to blame for her father leaving the family: *“Not only did he leave me and my sister when he divorced my mom, but now it’s like I wasn’t even good enough for him to stay around.*” This idea seemed to prolong the healing process. Danica continued,

As a kid I just knew that I wasn’t good enough for him to stick around. My little sister wasn’t good enough, my mom, my other siblings, you know. So that’s kind of what middle school was like.

In addition to the tragedy of the suicide and feeling left by their fathers, the participants also related how they were separated from family members from their father’s side of the family. Malcolm talked about how he is only in contact with his mother’s side of the family currently: *“My mom and my sisters, my maternal grandparents –that’s the constant is the maternal side.*” Justin reported similar experiences after the death of his father: “*I know there is a lot of anger and resentment in my family from both sides. Neither side of the family really gets along and I don’t see any of my father’s siblings anymore.*” At the time of the interview, Justin had not interacted with his father’s family in many years. Cody added that he too had experienced a separation from his father’s family. He reported not seeing or interacting with his uncles and grandparents on his father’s side. Cody said that over time, they *“separate[ed] themselves from us, it kind of grew apart.*”

#### 3.8.3. Acting Out

Several of the participants disclosed how their grief and pain from their father’s death served as a catalyst for maladaptive behaviors. Cody described a time of straying away from family values in order to find short-term relief from the pain, described as “self-medicating.” He shared his perspective:

I wasn’t a good boy for a long time, I strayed and that was primarily because of the pain that I had. I mean, it’s a lame excuse but it’s, it’s, it’s really true. Like, if, if you have all that pain, that’s the core problem if you don’t solve it. You can see all these other actions are made manifest because of that problem. And so, I think a lot of people that have had, had parents or loved ones commit suicide, they primarily start doing drugs, they do all these other things, it’s because they’re trying to self-medicate to cover up that pain. I did. For a long, long time and it was difficult to get out of.

Jesse talked at length about his ongoing battle with anger, depression, self-hatred, and seeking approval. He shared these perceptions:

I get my anger from him, I get my depression from him. I dislike myself and my life. I would be willing to sacrifice myself for a new me. I have that constant need of, at least, self-approval, not really approval from others, but, at least, I seek it from others. Does that make sense?... I constantly want to build myself up, but I don’t take anything from others. I’ll take their negatives but not the positives.

Delani described her self-blame for her father’s death:

Mom says that I was her ‘Mother’s Day present from hell.’ She says it jokingly, but apparently, I was the worst baby ever and I was born [the week of] Mother’s Day. I was extremely fussy. I cried all the time. I was just a terrible child. Mom says I was just, a huge stress, basically.

As Delani repeatedly heard this as a child, she worried that she caused her father to complete suicide, that she was the cause of her father’s death. She continued,

It makes me feel like maybe he did that [completed suicide] because of me because I was born. It took me so long to get over and my mom did not help with this part. Me hearing how difficult I was as a baby and then my dad committing suicide when I was so young, I was like, ‘Oh my gosh, I must have been a part of the reason, if not the whole reason,’ you know?

#### 3.8.4. Feeling Paralyzed, Conflicting Feelings, and Needing to Fix It

Several participants described feeling paralyzed after the suicide. Cody relates,

I remember that [cough] that since that day, for a long, long time, it felt like everything just stopped. Everything in my life, just… My progression, my happiness, like everything was just kind of off whack. Because there was that huge part of my life that was taken away in an instant.

Other participants shared how they felt stuck in their complex feelings that they had for their fathers. For example, Jesse described how he could not yet come to terms with the word “Dad.” He stated,

I can’t even call him Dad because there’s no such thing. So I don’t know what that means but… You know what I mean? The fact that everyone has a dad except me. I had a father, you know. That’s just biological, there’s nothing intimate whatsoever. Dad is more casual and whatnot. Father is just a clerical word. I want to say Dad, but I can’t. So even to this day I still have that mind war.

Danica also shared her complex, sometimes conflicting emotions towards her father:

People who commit suicide are so selfish and I thought that all the way up until adulthood. Which is a shame because there’s a lot more to suicide than that, but as a kid I just knew that I wasn’t good enough for him to stick around. I also hated being compared to my dad. Forever, I hated it. Being averse to him has made it much easier because I don’t like him as a person. I’m happy that he’s not my father. I’m happy that I wasn’t raised with him. But I hate that it’s like I wasn’t even good enough for him to stay around.

Justin shared how a Sunday School teacher’s impulsive and unthinking remark caused conflict in his feelings for his father. Justin related, *“As a little kid I worried that I would never see my dad again. So, one day in church I asked the Sunday School teacher,*
*‘If someone kills themselves can they go to heaven?’*” Without a second’s hesitation, the Sunday School teacher looked at Justin and replied. *“No.*” Justin explained, *“**He gave me a pretty frank answer and was like ‘well, it’s technically like a form of murder and so you can’t really go to [heaven],’ or whatever.*”

Cody also explained an ongoing conflict in his feelings about his father’s death. He related, “*They keep inviting me to speak in church on Father’s Day. To this day, I keep declining. I can’t do it.*”

In addition to feeling paralyzed and dealing with conflicting feelings about their fathers, almost all of the participants talked about needing to fix something related to their fathers. For example, although Malinda did not seem to blame herself for her father leaving, she did seem to indicate that it was her responsibility to learn about him and to get to know him. Malinda said,

It kind of makes me sad that I feel like I don’t know a lot about his life, like, I feel, like, I don’t, like, know him very well um, but, the thing is, is, like, I can still, like, make that happen now, you know.

Even when the participants attended therapy sessions, they gave the impression that each individual is responsible for “fixing it.” Danica discussed how even though counseling sessions were very confusing for her, she still felt she needed to “fix it.” She further explained:

Like, I couldn’t understand a metaphor to save my life. I just couldn’t connect those pieces, so when I was with the counselor and they’re trying to, you know, tell me the grass isn’t greener on the other side, like that made no sense to me at all. They were trying to teach me through proverbs. I had no idea what was going on because I was, just my brain didn’t function like that. So, I was never told, straight up, like how, you know, this is what’s going on, this is what you need to do to fix it.

Jesse also seemed to indicate that all of his years in counseling taught him he could fix things by “sucking it up.” He commented, “*I’ve always had some kind of counselor. And um, basically all you do is you can vent and then suck it up. That’s what I’ve basically learned in life.*”

### 3.9. Participants’ Perceptions of Books

Participants’ descriptions of their perceptions of helpful and unhelpful aspects of books are listed in Table 4. Participants reacted to the books in various ways by describing which books they felt would have been helpful or unhelpful to young CSoPS. It struck the researchers how closely participants’ reactions to the books seem to mirror their experiences of finding out about the suicide and how they reacted to the suicide.

#### 3.9.1. Characteristics of Books That Participants Found Helpful

Participants described books as helpful when the book appeared to be gentle; when the book seemed positive; when the book contained bright colorful images and illustrations; when the book helped them to feel connected to a loved one; when the book addressed fears; and when the book acknowledged the challenges of losing a parent to suicide.

**Gentle.** Cody delighted in the book *Are You Like Me?* because of its gentle way of easing children into the difficult subject or memory of a loved one’s suicide. He said, “*It’s like guiding them through. It is a good analogy. It is helpful. It eases into it very gently.*”

**Positive.** Malcolm also labeled *Are You Like Me?* as a “wonderful book.” He explained, “*It’s a positive book and helps make you think about things you weren’t thinking about.*” Jesse agreed. He expanded, “*So the pictures, are obviously big and colorful, kids like that. Good imagery. Yep, this book is perfect.*”

**Connecting and Attaching.** As a whole, *The Invisible String* stood out as a favorite among participants. Several commented that the book helped them remember that they were connected to loved ones always. Cody commented, “*This is a really good book. I liked it because my mom always said, ‘No matter where you are, no matter what you do, I will always love you.’*”

Delani also described how *The Invisible String* addressed the need to feel connected. She said, “*I like how the book talks about how kids would take the invisible string with them wherever they go to feel connected.*” Delani seemed deeply impacted by the mother in *The Invisible String* when she told the children that no matter how she feels, whether angry or not, their connection would not be broken. Delani continued, “*This book* [tapping on *The Invisible String*] *is something that kids will always remember. Yes, The Invisible String. Kids would connect to it.*”

**Addressed Fears.** Participants also seemed drawn to *The Invisible String* because the book addressed fears attached to their father’s suicide. Delani explained,

The book says, ‘Can a string reach all the way to Uncle Brian in heaven?’ That’s good. I would’ve enjoyed this book as a kid. When the book says, ‘does the string go away when you’re mad at us?’ The mother responds, ‘Never.’ I like that. Because that was my greatest fear with what happened to me as a little girl.

Delani was so taken with the idea that a book about suicide should address fears, that even though she found the book, *After a Suicide: An Activity Book for Grieving Kids*, to be unhelpful, she responded positively to the one line in the book that addressed a fear she had related to the suicide. She commented, “*This page is okay.*” She referred directly to one statement in *After a Suicide: An Activity Book for Grieving Kids*, which read: “*You may think you did something to cause this to happen, this is not true.*”

Delani also responded positively to the idea of the invisible string applying to separations caused by death and other traumatic situations. She expanded,

I felt like [*The Invisible String*] addressed a lot of things. Like it addressed death, it addressed the simple fears of a child…. Like their mother or father being angry at them, the string is still there even if the parents are angry at the child. That gets at a child’s main fear of someone not liking them or someone being mad at them.

Malcolm agreed: “*This* [*The Invisible String*] *is a nice book with a very happy, helpful message. It is applicable in all scenarios.*”

**Acknowledging.** Participants also felt that books that acknowledged the challenges of living with losing a loved one to suicide were helpful. Malinda explained,

I liked that it [*Not the End*] showed, like, that there is, like, dark time, you know. And I really liked that they showed the family, like, engaging in, like, positive, uplifting, happy activities, like, after the death.

Malinda was also drawn to another book that acknowledged the pain associated with suicide. She said,

I think that’s important to acknowledge, like, yeah, it’s painful, and I think just, like, shoving that away and, like, not acknowledging the pain, like, is not beneficial. Um, I really like this book, this part [*Samantha Jane’s Missing Smile*] where she says, ‘But sometimes I worry that if I talk to you about dad, you’ll start to feel sad.’ So I like that this book acknowledges that and says that it’s okay and that, like, the parent figure, like, wants the child, um, to talk to them.

#### 3.9.2. Characteristics of Books Participants Found Unhelpful

Participants seemed to be specifically sensitive to books that seemed confusing. They reported that these types of books would not be helpful to children. Participants also described books as unhelpful if the book seemed to be rough or insensitive, had perceived missing parts, were cold, were triggering, were negative in nature, or if the book seemed to increase fear.

**Confusing.** Danica described the book, *Luna’s Red Hat*, as unhelpful because she found it to be confusing. She said, “*I don’t think I would’ve understood this book as a kid. I would’ve been like, okay what’s going on? I don’t understand. I don’t even know if I would get it.*”

Malinda discussed how she found the book, *Not the End*, to be confusing. She stated, *“I don’t know if I got the analogy of, like… like, your life is a book and, like, turning the pages just because I think the process is more of a continuum rather than, like, chapters and segments.*”

Delani also shared how children would find the book, *After a Suicide: An Activity Book for Grieving Kids*, confusing. She said, “*It would be hard for a kid to relate to this. I don’t get this.*”

Although some participants found the book, *Are You Like Me?*, to be helpful, Jesse had a strong negative reaction to the book, which he found confusing. He stated,

The book is too nuanced. You need to say ‘to stop them from living’ or something like that because like, if you’re just like ‘stop working’ it’s confusing. It’s like, no, you need to be more straightforward. I just don’t like that phrasing. There’s got to be a better way of saying that. I would put that first, ‘be sure to talk to grownups about your fears’ because this one kind of seems backward. It needs to be near the beginning.

**Insensitive.** Cody found the book, *My Uncle Keith*, to be unhelpful. He explained, “*I don’t think it’s helpful because it’s not, uh, explained with a sensitivity.*”

**Missing Parts.** Participants also reacted negatively and found books unhelpful that seemed to have obvious omissions and “missing parts”: *I’m basically just seeing friends and family [in* Are You Like Me?*]. Is there ‘hey, you can also refer to someone at school’? I’m not seeing that.*

**Cold.** Books that appeared cold received negative reviews and were rated as unhelpful by the participants. For example, Delani believed that the cold language appearing in the book, *After a Suicide: An Activity Book for Grieving Kids*, would hinder a young child’s understanding and hamper their ability to connect with the book’s information. She said, “*I don’t really like this book. It’s just, cold, not creative. I shouldn’t be so critical, but it is just cold. It’s not a story.*”

**Triggering.** As soon as Justin saw the cover of the book, *Little Flower Bulb*, he immediately reacted negatively, “*I don’t like the art style, right away.*” Participants’ overall feelings based off of the illustrations was quite negative. Justin (whose father died by hanging) commented, *“See his head is kinked. And now, we’ve been talking about my dad so much, it makes it look like he’s hanging.*” The cover illustration of the *Little Flower Bulb* includes a little boy whose head is tilted to one side. This was particularly off-putting and triggering for Justin, who commented, “*He kind of looks dead right there, to be honest, like, that hollow look in his eyes.*”

Jesse described a book that seemed to be triggering for him as well. He commented, “*The thing is, this kid’s kind of screwed [in the book,* Are You Like Me?*]. It makes my brain feel like bullshit!*” Dana also described a triggering moment after reading the book, *Luna’s Red Hat.* Wistfully, she commented, *“And [this book] just made me feel so sad.*”

**Increases Fear.** In addition to being triggering, books that seemed to induce fear were rated as unhelpful. Justin related his negative impression of the *Little Flower Bulb*, “*Maybe it’s intended that way to, like, you know, let kids know that life can be scary and dark and rough and stuff like that. But this is not a book for children.*”

## 4. Discussion

Prior research and feedback from mental health practitioners parallel our participants’ expressed needs and concerns. Following their father’s suicide and across the ensuing years, participants described limited opportunities to talk about their father’s life and death [5,6,18,20,26,35]. Children were confused about *how* their father died. After discovering how he died, they also wanted to know *why* he chose to complete suicide. For young children, these are very important questions [75]. Participants reported that adults assumed that young children could not understand what was going on around them. Participants sensed the adults’ intensity of emotions but were confused about how these strong emotions connected with what was happening. Very young children rely on the adults around them to help them build memories of their deceased parent.

Several participants remembered feeling guilty about possibly being the cause of their father’s suicide. As the children reviewed the books, the fears and confusion discussed in the interviews were reflected in their comments. Based on participants’ expressed emotional needs, books shared with young CSoPS should assist in opening conversation, clarifying the swirling confusion in their young lives, providing reassurance that they are loved unconditionally, and reassuring them that the suicide was not their fault.

### 4.1. Postvention Support: Bibliotherapy

In Regher et al.’s study, participants—mental health paraprofessionals who worked with child survivors—noted the usefulness of children’s books to open communication about suicide [6]. In our study, we reviewed input from CSoPS. Their comments made during the interviews and while reviewing the children’s books described challenges and expressed needs that aligned with Worden’s four tasks of grief [14]. Although each person’s grief is unique, generalities across experiences provide a structure for offering bibliotherapy as part of postvention support. A handout for parents/caregivers might include children’s books that align with one or more of the tasks of grief: (a) understanding and accepting the reality of death, including how he died and why he chose to complete suicide; (b) facing the pain of grief, not in isolation but with a caring supportive network, including addressing guilt the child may be harboring; (c) adjusting to the life after a loved one’s death, particularly addressing potentially strained relationships with the father’s family, preparing for events such as moving, and living without one’s father; and (d) remembering the life and death of the deceased person, which may include fostering a connection with the deceased loved one, honoring the father’s life by visiting the grave site, and creating a memory book of photos and cherished keepsakes. Caregivers need to help very young children build positive memories of their deceased parent. We consider the last mentioned task to be particularly challenging for the youngest of CSoPS because they lacked opportunities to build a firm connection/attachment with their father.

The young CSoPS in this study experienced difficulty when facing the challenges associated with their father’s suicide. Because the facts about the parent’s death are often not shared with the young child, the CSoPS will have difficulty accepting death and understanding that death is real. Some children may not see the deceased body and may not attend the funeral. In some situations, the father’s body may be cremated due to circumstances related to the suicide. Further blunting the child’s ability to accept and understand the parent’s death is the heavy stigma associated with suicide [4,5,26]. Postvention support should target destigmatizing suicide and opening conversation that supports and validates challenges associated with survivors’ experiences. Caregivers also need talking points and examples of how to talk about suicide with young children in a developmentally appropriate way. Caregivers may not know what to say or how to offer support. In regard to CSoPS, Montgomery and Coale provide a booklet that gives examples of what to say and how to respond [35].

We recommend matching books with the child’s specific situation, possibly offering a few books and letting the child select one that they prefer. Initially, books may not be specifically about a parent’s death, but about expressing emotions, about grief and loss in general, and then moving into suicide-themed books as the child is ready to talk specifically about suicide. Families and school professionals may want to shield children from emotional pain, but we all must learn to live with our grief.

CSoPS need support when adapting to changes in their life after the father’s death. As mentioned by several participants, family dynamics changed overnight, with the father’s side of the family often ceasing to have contact with the children or surviving parent. Financial hardships may necessitate the mother working or remarrying someone who can support her and the children. These changes are substantial. The surviving parent is often emotionally unavailable to the children due to the trauma and personal crisis they are facing [76]. CSoPS need to be reassured that they are loved and that family members, caring adults, teachers, and counselors will support them and their mother during this difficult time. Children also need to know that their surviving mother will get the help and support she needs. The emotional stability and resilience of the surviving parent is of critical importance to child survivors [17,76].

Children need to memorialize their deceased father, building connections that will be sustained across time. Cerel and Sanford indicate that “It’s not who you know, it’s how you think you know them” [77] (p. 76). They noted that for CSoPS, the actual facts and the formally identified relationship do not drive how children respond and cope with the reality of the suicide. They indicate that children’s memories and grief are grounded in the “nature and perceptions of the relationship” [77] (p. 76). Older children will have a more established relationship with the deceased parent, but for very young children this is often missing. The memories participants reported were often molded by those around them. Over time, others’ stories became the child’s memories. In some cases, these stories were painful and may need to be reframed and retold. In bolstering the strength to sustain their children’s emotional wellbeing, surviving parents must be careful to avoid portraying children in a negative way that raises guilt and increases the likelihood of children questioning their own responsibility in their parent’s suicide.

Participants in this study were especially impressed with *The Invisible String*. Making connections that cannot break and are always present—regardless of what we do, what we say, or where we go—addressed a critical emotional need in our participants. Reassuring children that they are connected to their family and to their deceased father would be helpful when addressing the need to memorialize the deceased.

### 4.2. Limitations

In this study, we included seven young child survivors who experienced their father’s suicide. This is a narrow selection of participants. However, in this study the close match between the study phenomena and the participants’ life experiences increased the information power [66]. Additionally, carefully planned and executed in-depth interviewing accompanied by careful member checking provided the opportunity to gather and analyze rich, thick, multidimensional descriptions of lived experiences [60,71]. Malterud et al. note that sample specificity, quality of dialogue, and intentional use of analysis strategy correspond with smaller sample sizes that provide sufficient information power [66]. This study’s methodology and data analyses allowed an in-depth exploration of life experiences [66].

Additionally, this study’s findings may not generalize to other subgroups of CSoPS, such as child survivors of a mother’s suicide, and our findings may not generalize to older children. Another limitation is that we did not investigate response differences between female and male survivors. Furthermore, this study only investigated participants’ perceptions of a small selection of children’s books and did not investigate the actual effectiveness of children’s books in addressing the needs of CSoPS.

Although this study’s data are retrospective and reflect participants’ memories of their experiences following their parent’s suicide, regardless of the accuracy of children’s memories, their perceptions are these survivors’ reality [53,54,55]. We ascribed to the idea that retrospective interviews by adults have been found to include more depth and emotional poignancy than memory work with children and can forge a missing voice for vulnerable or silenced populations, such as CSoPS [53,54,55]. As such, we attended to recognized principles of rigor for retrospective work by developing research questions that focused on a specific situation (parent suicide) rather than a life in its entirety, and compared present thoughts (which books may be beneficial) about past events (to child survivors of parent suicide). As a team, through rounds of reflexivity, we realized that, regarding treatment, child survivors’ perceptions across time, whether consistent or inconsistent, provided targets for supportive intervention when addressing the postvention needs of CSoPS [78].

### 4.3. Recommendations for Future Research

Suggestions for future research may include gathering information from those who work with very young CSoPS, identifying their perceptions of which interventions are most effective in opening communication about the parent’s suicide. For example, gathering information from multiple sources, including the surviving parent, caregivers, and teachers, would also assist in clarifying which interventions may be most helpful. Additionally, longitudinal studies, though more challenging to conduct, would provide insights into the effects of suicide prevention efforts to lower suicide in CSoPS. Furthermore, longitudinal studies would offer the dynamic effects of children’s developmental maturation and change in evolving grief and perceptions of coping across time.

## 5. Conclusions

The importance of investigating how best to support young CSoPS should not be underestimated. Supporting CSoPS not only addresses postvention needs but aligns with suicide prevention. These young children face an uphill battle in mastering Worden’s tasks of grief, which are complicated by minimal communication with family members and other caring adults [14]. Very young CSoPS must be supported in asking tough questions about their parent’s suicide. Additionally, very young children need assistance in forming positive memories of their deceased parent that encourages a healthy attachment.

Surviving parents may need support in managing their own personal needs while also providing adequate support to their children [76,79]. We recommend that professionals provide information to help parents carefully select children’s books to address the child’s specific needs. Children’s needs may be organized by Worden’s tasks of grief [14].

## Figures and Tables

**Table 1 ijerph-18-11384-t001:** Participant Demographics.

*Name*	Participant’s Gender	Participant’s Age at the Time of Father’s Suicide	Mode of Father’s Suicide
*Danica*	F	5 years old	Overdose
*Malcolm*	M	3 years old	Poisoning in vehicle
*Jesse*	M	3 months	Gunshot wound
*Delani*	F	1 year old	Hanging
*Malinda*	F	4 years old	Poisoning in hotel room
*Justin*	M	5 years old	Hanging
*Cody*	M	5 years old	Gunshot wound

NOTE: All names are pseudonyms.

**Table 2 ijerph-18-11384-t002:** Description of Children’s Books Shared with Participants.

Title	Author	Synopsis
*After a Suicide: A Workbook for Grieving Kids*	The Dougy Center, The National Center for Grieving Children & Families	This book includes activities and prompts for conversations to help a child understand and cope with a loved one’s suicide. Includes quotes from children, some specific to a parent’s suicide. Brightly colored pages with illustrations drawn by children.
*Are You Like Me?*	Miki Tesh and Eric Schleich	This book provides a starting point and safe place to start processing grief after a loved one dies by suicide. This book is not specific to a parent’s suicide. Covers a wide range of feelings associated with a suicide death.
*The Little Flower Bulb*	Eleanor Gormally	The main character is a little boy whose father died by suicide. The family learns to cope with the pain of their father’s death. In his memory they plant a flower bulb and care for it, awaiting its bloom after winter ends. The illustration style may be perceived by some as *eerie.*
*The Invisible String*	Patrice Karst	This book does not specifically address parent suicide. The three main characters are a mother and her young daughter and son. To calm her children’s fears, the mother describes the love between her and them as an invisible string. Regardless of the situation, this invisible string between those you love and yourself stays connected. Illustrations are simplistic drawings without a lot of detail.
*Luna’s Red Hat*	Emma Smid	Luna, the main character, talks with her father about why her mother died by suicide. Mental illness is described in child-friendly language with depression illustrated as a tangled dark cloud hanging over the mother’s head. As they discuss her mother’s death, Luna expresses strong feelings. After venting her feelings, the story ends with Luna and her father sharing pleasant family memories that include her mother.
*When Someone Very Special Dies: Children Can Learn to Cope With Grief*	Marge Heegoard	This informative workbook leads children through different drawing/writing exercises, helping children cope with the reality of death and associated grief (not suicide specific). Prompts encourage children to write inforation, color pictures, and draw pictures in the workbook.
*Not the End: A Child’s Journey Through Grief*	Mar Dombkowski	This is based on a true story. The little girl, who is the main character, tells the reader all about how her family has grown and how life continued after her father died. The exact cause of death is not specified. This book is not suicide specific.
*Samantha Jane’s Missing Smile*	Julie Kaplow and Donna Pincus	The story centers on a conversation between Samantha Jane and her neighbor, Mrs. Cooper. Before her father died, Samantha Jane was happy. Samantha Jane starts to feel better as she talks to her neighbor. This book is not suicide specific but emphasizes the importance of processing thoughts and feelings and not keeping grief submerged. The story uses a metaphor of a stick pushed under water which, when released, pops to the surface.
*My Uncle Keith*	Carol Ann Loehr	This book is framed around a conversation between a young boy, Cody, and his mother. They are talking about Uncle Keith’s suicide. They talk about Uncle Keith’s mental health and the need to seek help for mental health issues, including depression.
*Tear Soup: A Recipe for Healing After Loss*	Pat Schwiebert and Chuck DeKlyen	The main character is an older woman, Grandy. Her husband is deceased. As an expression of her grief, she cooks “tear soup.” This book considers the nature of grief and how people deal with grief in different ways across time. This book is not specific to suicide.

**Table 3 ijerph-18-11384-t003:** Sample Participant Portrait, How the Participant Found out About the Father’s Suicide (Cody, age five).

Emotion Codes	Process Codes
Confusion [participant] Um… so I was five years old, it was November…when was it? It was a while back. So anyway, so my dad hadn’t come home for a little, for a couple days or a day or two, I can’t remember for how long it was. Confusion [participant] They just came to our house [grandparents and uncles suddenly show up] and I thought that was kind of weird. Distraught [uncle] and he was visibly distraught. Confusion [participant] I didn’t know what was going on. Confusion [participant] And so later on, I don’t know what age or how old I was when I found out what really happened. Confusion [participant] but… I found out that he committed suicide… Well, I didn’t find out the method. I just know he committed suicide. Confusion [participant] I didn’t find out the method until a while later. I don’t member how old I was… The way I found out that he shot himself was through his death certificate.	Others missing [Dad] So anyway, so my dad hadn’t come home for a little, for a couple days or a day or two, I can’t remember for how long it was. Others showing up [unexpected family members] But my grandparents suddenly showed up and a couple of my uncles showed up as well. Playing [participant] I was young, I was playing in the gutter with some toys or whatever. Others talking [uncle to participant] And… my uncle Josh, he came up to me and he talked… he started talking to me. Others dsregulating [uncle’s emotional dysregulation] and he was visibly distraught. Asking [participant to uncle] I asked him, I said, ‘What’s wrong?’ Hearing [participant] And then, in the background, I heard my mom just collapse and she was screaming. Others Collapsing [mother] I heard my mom just collapse and she was screaming. Others Screaming [mother] I heard my mom just collapse and she was screaming. Others talking [grandparents] grandparents had informed her what had happened. Others talking [uncle] And then my uncle told me, like, ‘Hey, your dad passed away.’ Others showing up [uncle] Like I said, my uncle, he came and… He showed up. Separating [participant] my uncle came to me separately, Others talking [uncle to participant] one of my uncles, and he started talking to me about it, and he said, ‘Hey, your father passed away.’ Others withholding they didn’t say how it happened, what happened, anything like that.

**Table 4 ijerph-18-11384-t004:** Participants’ Description of Books as Helpful or Unhelpful: Process Codes and Emotion Codes.

Participants Perceived Aspects of These Books to Be “Helpful”	Process Codes	Emotion Codes
Are You Like Me? *The Invisible String* *Not the End* *Samantha Jane’s Missing Smile*	Approving Guiding Gradual easing into Helping Approaching gently Relating Attaching Liking Understanding Applying Connecting Addressing Encouraging Appreciating Representing Acknowledging	Happy Nice Love Positive Hopeful
**Participants Perceived Aspects of These Books to be “Unhelpful”**	**Process Codes**	**Emotion Codes**
*My Uncle Keith* *After a Suicide: An Activity Book for Grieving Kids* *Are You Like Me?* *Luna’s Red Hat* *Little Flower Bulb*	Leaving Triggering Increasing Fear Confusing Missing parts	Fear Insensitive Confused Sad Ambivalent Unclear Unhelpful Cold Negative

Note: see Table 2 for a complete description of the books.

## Data Availability

Data presented in this study are available on request from the corresponding author. The data are not publicly available due to IRB protocols.

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
