# Peer review of "Very Young Child Survivors’ Perceptions of Their Father’s Suicide: Exploring Bibliotherapy as Postvention Support"

_ijerph, 2021, doi:10.3390/ijerph182111384_

Round 1
Reviewer 1 Report
This is an important and interesting study, but the integration between the methods should be elaborated and comparison with ref 7 is relevant.

Reviewer 2 Report
This is an important topic, and I applaud the authors for their efforts in cultivating therapeutic programming for this vulnerable group. For me, this paper is incomplete: One would wish for the data presented to inform a program for children who have lost their parents to suicide to see if the themes captured by the focus group apply to (and help) children experiencing loss in the present. The sample is too small, retrospective, and confounded with memory bias to meet the authors' aim of developing a program targeting current pediatric survivors of loss. They are asking adult survivors to look back in time and speculate, "Would this book have been helpful to my younger self in the immediate aftermath of my loss?" -- helpful but insufficient to answer the question of what would be helpful to pediatric survivors in the present. If possible, I would be very interested in reviewing a submission describing the pilot of the bibliotherapy program created based on the learnings of this Delphi, but otherwise, I wish the authors well in their endeavors, because, as I say, they are embarking on important work -- just not yet ready for publication.
Round 2
Reviewer 2 Report
Thank you for addressing the reviewers' comments. As there has been no therapeutic program developed yet, and as the exploration of these books appears to be in its very initial phase (an Amazon pull with consultation), I strongly recommend a change of title to minimize the impression that there has been an intervention. Perhaps: "Very Young Child Survivors’ Perceptions of Their Father’s Suicide: Exploring Bibliotherapy as Postvention Support" or "Very Young Child Survivors’ Perceptions of Their Father’s Suicide: A Small-Sample Exploratory Study". I would also recommend that the authors look to Magination Press's resources before making claims about the novelty of their idea and include this resource in their reference list: https://www.maginationpressfamily.org/stress-anxiety-in-kids/tag/grief/
Also, what were the qualifications of the "bibliotherapy experts" referenced in the authors' response to reviewers? This information should be included in the manuscript as well.
Author Response
Thank you for the opportunity to respond to further reviewer feedback. Please see the attachment.
